

# Preventive effects of *Salvia officinalis* leaf extract on insulin resistance and inflammation in a model of high fat diet-induced obesity in mice that responds to rosiglitazone

Mohamed R. Ben Khedher[1], Mohamed Hammami[1], Jonathan R.S. Arch[2], David C. Hislop[2], Dominic Eze[3], Edward T. Wargent[2], Małgorzata A. Kępczyńska[2] and Mohamed S. Zaibi[2]

[1] Biochemistry Department, Research Laboratory 'Nutrition—Functional Food & Vascular Health', Faculty of Medicine, University of Monastir, Monastir, Tunisia
[2] Buckingham Institute for Translational Medicine (BITM), Clore Laboratory, University of Buckingham, Buckingham, United Kingdom
[3] Medical School, University of Buckingham, Buckingham, United Kingdom

Corresponding author
Mohamed S. Zaibi,
mohamed.zaibi@buckingham.ac.uk

## ABSTRACT

**Background**. *Salvia officinalis* (sage) is a native plant to the Mediterranean region and has been used for a long time in traditional medicine for various diseases. We investigated possible anti-diabetic, anti-inflammatory and anti-obesity effects of sage methanol (MetOH) extract in a nutritional mouse model of obesity, inflammation and insulin resistance, as well as its effects on lipolysis and lipogenesis in 3T3-L1 cells.

**Methods**. Diet-induced obese (DIO) mice were treated for five weeks with sage methanol extract (100 and 400 mg kg$^{-1}$/day bid), or rosiglitazone (3 mg kg$^{-1}$/day bid), as a positive control. Energy expenditure, food intake, body weight, fat mass, liver glycogen and lipid content were evaluated. Blood glucose, and plasma levels of insulin, lipids leptin and pro- and anti-inflammatory cytokines were measured throughout the experiment. The effects of sage MetOH extract on lipolysis and lipogenesis were tested *in vitro* in 3T3-L1 cells.

**Results**. After two weeks of treatment, the lower dose of sage MetOH extract decreased blood glucose and plasma insulin levels during an oral glucose tolerance test (OGTT). An insulin tolerance test (ITT), performed at day 29 confirmed that sage improved insulin sensitivity. Groups treated with low dose sage and rosiglitazone showed very similar effects on OGTT and ITT. Sage also improved HOMA-IR, triglycerides and NEFA. Treatment with the low dose increased the plasma levels of the anti-inflammatory cytokines IL-2, IL-4 and IL-10 and reduced the plasma level of the pro-inflammatory cytokines IL-12, TNF-α, and KC/GRO. The GC analysis revealed the presence of two PPARs agonist in sage MetOH extract. *In vitro*, the extract reduced in a dose-related manner the accumulation of lipid droplets; however no effect on lipolysis was observed.

**Conclusions**. Sage MetOH extract at low dose exhibits similar effects to rosiglitazone. It improves insulin sensitivity, inhibits lipogenesis in adipocytes and reduces inflammation as judged by plasma cytokines. Sage presents an alternative to pharmaceuticals for the treatment of diabetes and associated inflammation.

## INTRODUCTION

In the recent decades, there has been renewed interest in traditional and alternative medicine and thousands of potential medicinal plants have been screened to identify bio-active lead components. *Salvia officinalis* (Sage) has been extensively used as a medicinal plant in treating several diseases and recent studies have shown promising activity in treating cancer (*Shahneh et al., 2013*), heart disease, dementia and obesity (*Hamidpour et al., 2014*).

Studies have suggested that sage extracts enhance glycemic balance in normal and diabetic animals. *Alarcon-Aguilar et al. (2002)* showed that a water ethanolic extract from *S. officinalis* injected intraperitoneally had hypoglycemic effects in fasted normoglycemic mice and in fasted alloxan-induced mildly diabetic mice. In addition, *Eidi, Eidi & Zamanizadeh (2005)* showed that sage methanolic (MetOH) extract given intraperitoneally reduced significantly serum glucose in fasted streptozotocin (STZ)-induced diabetic rats without changes in plasma insulin levels. In another study, sage ethanolic extract significantly decreased serum glucose, triglycerides and total cholesterol, whereas it increased serum insulin levels in STZ-treated diabetic rats as compared with control diabetic rats (*Eidi & Eidi, 2009*). Sage essential oil tested in normal and in alloxan-induced diabetic rats improved glycemia (*Baricevic & Bartol, 2000*) and increased the response of the hepatocytes to insulin in normal animals but not in hepatocytes isolated from STZ diabetic rat (*Lima et al., 2006*). Sage is reported to elicit antidiabetic effects largely due to activation of peroxisome proliferator-activated receptors (PPARs) (*Christensen et al., 2010*).

Most of the studies described above have investigated the anti-diabetic effects of sage in alloxan- or streptozotocin-induced diabetic animals. However, the effects of sage on insulin sensitivity and glucose tolerance in a nutritional animal model of obesity and insulin resistance have not been described before. The aim of our present study is to assess the potential anti-inflammatory, anti-obesity, and anti-diabetic effects of low and high doses of a MetOH extract of *S. officinalis* leaves, in a high fat diet-induced obesity mice model, which is a nutritional animal model of obesity associated with dyslipidemia, inflammation and insulin resistance and to appraise the effect of sage MetOH extract in 3T3-L1 cells on lipolysis and lipogenesis.

## MATERIALS AND METHODS

### Chemicals and reagents

Methanol (Sigma-Aldrich, Munich, Germany), dimethyl sulfoxide (DMSO, Biotech grade, 99.98%; Sigma–Aldrich, Munich, Germany), Dulbecco's modified Eagle's medium (DMEM), 0.25% trypsin-EDTA (1X), fetal bovine serum (FBS), streptomycin/penicillin (Gibco BRL, Life Technologies, Carlsbad, CA, USA), bovine insulin (Sigma I-5500), dexamethasone, (Sigma D-4902), 3–isobutyl–1 methylxanthine (IBMX; Sigma I-7018), rosiglitazone maleate (SRP0135r; Sequoia RP, UK).

## Preparation of plant material

Leaves of *Salvia officinalis* (*Lamiaceae* Plant family) were collected from the open field botanic garden of the Higher Institute of Agronomy, University of Sousse, Tunisia and were identified by Pr. Rabiaa Hawéla at the cited institute. Voucher specimens were deposited at the Faculty of Medicine of Monastir, Tunisia, and referenced as SO011. Air dried leaves were submitted to extraction with 80% MetOH solution in a Soxhlet apparatus for 24 h. The solvent was then filtered and evaporated by Rotavapor at 55 °C. The recuperated aqueous portion was lyophilized and stored at −20 °C, for fatty acids (FAs) analysis, and for *in vitro* and *in vivo* experiments.

## Fatty acid methylation and analysis

Fatty acid (FA) extraction was performed using a modified method of *Folch, Lees & Sloane Stanley (1957)*. Heptadecanoic acid (C17:0) was used as an internal standard in order to quantify FAs. Total FAs were converted into their methyl esters using MetOH/$H_2SO_4$ at 2.5%. FA methyl esters (FAMEs) were analyzed using a Hewlett Packard 5890 IIGC (Agilent Technologies, USA) equipped with Flame Ionization Detector (FID) and Supelcowax$^{TM}$ 10 capillary column (30 m × 0.32 mm, *i.d.*, 0.25 μm film thickness) with a stationary phase made of polyethylene glycol. FAMEs were identified by comparing each sample with a standard FAME reference mixture. FA acid peak areas were calculated using HP ChemStation software, quantified according to their percentage area and expressed in μg/g of dry plant material.

## HPLC Phenolic compounds analysis in sage methanol extract

HPLC analysis was carried out as reported by *Mhamdi et al. (2009)*. Separation was performed using a Hewlett-Packard liquid chromatographic system (Waldbronn) equipped with a Rheodyne Model 7125 injector (Rheodyne LP Cotati, CA, USA), a HP 1100 pump system and an HP 1040M detector (diode array detection system, DAD). The column was an XDB-C18 (250 mm × 4.6 mm I.D., 5 μm particle size). The mobile phase consisted of acetonitrile/acidified water (solvent A) with 0.2% sulfuric acid (solvent B). The selected gradient was as follows: 15% A/85% B, 0–12 min; 40% A/60% B, 12–14 min; 60% A/40% B, 14–18 min; 80% A/20% B, 18–20 min; 90% A/10% B, 20–24 min; 100% A, 24–28 min (*Bourgou et al., 2008*). The flow rate was maintained at 2 mL/min and a final volume of 20 μL was injected. The data were stored and processed by an HPLC Chemstation (Dos Series; Hewlett-Packard, Palo Alto, CA, USA) and identification of compounds was carried out by comparison of their retention time with respect to pure standards analyzed under the same operating conditions.

## *In vitro* experiment
### Cell culture

3T3-L1 cell line was purchased from Sigma, UK (Ref: 86052701). After a few passages in the growth medium containing basal Dulbecco's Modified Eagle's Medium—high glucose, (DMEM) supplemented with fetal bovine serum (FBS) 10%, and Penicillin-Streptomycin (S/P) 100 IU/ml, the cells were seeded in 24 well plates in the growth medium until they reached confluency. To initiate pre-adipocytes differentiation into adipocyte-like

cells, the cells were incubated in the differentiation medium (day 0) containing 10% FBS supplemented DMEM with S/P, 5 μg/ml bovine insulin, 0.5 mM IBMX and 0.5 μM dexamethasone. On day 3, the differentiation medium was replaced by the nutrition medium (10% FBS supplemented DMEM with S/P and $5.10^{-3}$ mg/ml bovine insulin). By day 7–10, mature adipocytes were obtained.

### Glycerol release

After complete differentiation of pre-adipocytes, the nutrition media was removed and mature 3T3-L1 cells were incubated for 90 min at 37 °C, in 0.5 ml of DMEM/ Ham's F-12[1:1], containing 0.1% of BSA and with or without 0.2, 1, 5, 25 and 50 μg/ml of sage MetOH extract dissolved in 0.01% DMSO. The glycerol released into the medium was quantified by using a colorimetric method (Glycerol kit, GY105; Randox, Crumlin, Dublin, Ireland Randox). Optical density was measured at 520 nm using SpectraMax 96- well plate reader.and results expressed in μmol/mg of cell protein.

### Lipid droplets accumulation

To evaluate the effects of sage extract on lipid droplets accumulation, the cells were treated with the plant extract dissolved in 0.01% DMSO at the following concentrations: 0, 0.2, 1, 5, 25 and 50 μg/ml. The extract was added to the differentiation medium, the nutrition medium, or both media. Lipid droplets accumulation was assessed by staining lipids with Oil red O as employed by *Ramírez-Zacarías, Castro-Muñozledo & Kuri-Harcuch (1992)*. The absorbance of the eluted dye was measured at 500 nm,

## In vivo experiment
### Animal model

32 male mice (C57Bl6) aged 6–7 weeks on arrival (Source: Harlan UK) were fed on a high fat diet (60% fat by energy value; cat #D12492, Research Diets, US. Diet composition is described in Table 1) for 9 weeks before treatment. The mice were housed at 25 ± 1 °C, 40% to 60% air humidity and 12 h light/dark cycle in groups of four per cage with free access to high fat diet (HFD) and tap water. Animal experiments were conducted in accordance with ethical procedures and policies approved by the UK Government Animal Act 1986 (Scientific procedures) Project Licence (PPL 70/7189), entitled "Identifying diabetes, obesity and metabolic diseases therapeutics", and Animal Welfare and Ethical Review Board (AWERB) of the University of Buckingham, UK.

After 9 weeks of HFD feeding, mice were fasted for 5 h from 09:00 and fasting blood glucose and plasma insulin were measured. 31 mice exhibited plasma insulin levels over 250 pmol $l^{-1}$, one mouse was excluded because of normal plasma insulin level (less than 150 pmol $l^{-1}$). Mice were then allocated into four groups with approximately the same body weight, plasma insulin and blood glucose levels. Seven mice in group A, eight mice in group B, eight mice in group C and eight mice in group D.

**Table 1  Composition of the Rodent Diet with 60% kcal%fat.** The high fat diet was formulated by EA Ulman, Ph.D. (Research Diets, Inc., New Brunswick, NJ, USA). The respective fat percentages in saturated, monounsaturated and polyunsaturated fatty acids are 37.1%, 46% and 16.9%.

| Product # | High fat diet D12492 | |
| --- | --- | --- |
| | gm% | kcal% |
| Protein | 26.2 | 20 |
| Carbohydrate | 26.3 | 20 |
| Fat | 34.9 | 60 |
| Total | | 100 |
| kcal/gm | 5.24 | |

| Ingredient | gm | kcal |
| --- | --- | --- |
| Casein, 80 Mesh | 200 | 800 |
| L-Cysine | 3 | 12 |
| Corn Starch | 0 | 0 |
| Maltodextrin 10 | 125 | 500 |
| Sucrose | 63.8 | 275.2 |
| Cellulose, BW200 | 50 | 0 |
| Soybean Oil | 25 | 225 |
| Lard | 245 | 2,205 |
| Mineral Mix, S10026 | 10 | 0 |
| DiCalcium Phosphate | 13 | 0 |
| Calcium Carbonate | 5.5 | 0 |
| Potassium Citrate, 1 H2O | 16.5 | 0 |
| Vitamin Mix, V10001 | 10 | 40 |
| Choline Bitartrate | 2 | 0 |
| FD&C Blue Dye #1 | 0.05 | 0 |
| Total | 773.85 | 4,057 |

Animals were treated as described below for five weeks:

A: Control (water, 10 ml kg$^{-1}$/day bid)

B: Sage extract (100 mg kg$^{-1}$/day bid)

C: Sage extract (400 mg kg$^{-1}$/day bid)

D: Rosiglitazone (3 mg kg$^{-1}$/day bid)

The animals were dosed twice per day at 9:00 am and 5:00 pm.

## Analysis

### Biochemical analysis

Blood samples were centrifuged at 5,000 g for 6 min and plasma were collected into 96-well plates and stored at $-20\,°C$ until assayed. The tested biochemical parameters were: insulin (Ultra Sensitive Mouse Insulin ELISA kit, Catalog #: 90080; Crystal Chem, Downers Grove, IL, USA), leptin (Catalog #: 90030; Chrystal Chem), total cholesterol (ref: CH200; Randox Laboratories), triglycerides (ref: TR1697; Randox Laboratories), high density lipoprotein cholesterol (HDL-C) (ref: 354LB; Trinity Biotech, Bray, Co. Wicklow,

Ireland). Blood glucose was measured using a glucose oxidase reagent kit (Gluc-PAP, GL2623; Randox). All parameters were analyzed automatically using SpectraMax 250 and SoftMax Pro software.

### Body composition
Lean and fat mass were measured by Nuclear Magnetic Resonance (NMR), using the minispec LF 90$_{II}$ device (Bruker UK Limited). The different types of tissue were identified according to their density by comparison with a calibrated standard. Dedicated software is used to quantify amounts of lean and fat.

### Energy expenditure
After 35 days of treatment, energy expenditure was measured by indirect calorimetry as reported by *Stocker et al. (2007)*. Energy expenditure was evaluated based on the equation of Weir using customized software (*Arch et al., 2006*).

### Oral Glucose Tolerance Test (OGTT) and Insulin Tolerance Test (ITT)
OGTT and ITT were performed respectively at two and four weeks following the start of treatment with the plant extract. Mice were fasted for 5 h before the OGTT and ITT. We and others routinely use a fast of about this length. Fasting mice for 5–6 h instead of overnight might offer a better comparison to humans. Indeed, after having reviewed a number of studies, *Jensen et al. (2013)* concluded that because the mouse has a nocturnal circadian rhythm (two-thirds of total food intake are consumed during the night) and a higher metabolic rate, therefore, a 16–18 h fast affects metabolism far more in mice than in humans. Overnight fast is a routine event in humans, but not in mice.

For OGTT, 5 h-fasted mice received by oral gavage a glucose solution (in distilled water) 2.5 g/10 ml kg$^{-1}$, and blood samples (10 μl) were collected by incision from the tail for glucose measurement at 30 and 0 min before, and, 30, 60, 120 and 180 min after the glucose load. Plasma insulin levels were measured at −30 and +30 min. Blood glucose levels were measured using glucose oxidase reagent. For the ITT, 5 h-fasted mice received 0.75 IU/kg of insulin solution (Actrapid. HM, Novo Nordisk, Denmark) by intraperitoneal injection, and blood glucose was measured at just prior to the insulin injection, and at 10, 20, 30, 45, and 60 min following the injection.

### Liver Triglycerides measurement
About 150 to 300 mg of liver samples were used to assess triglycerides content as described by *Harzallah et al. (2016)*. The reading was performed using the Randox triglycerides kit.

### Liver glycogen measurement
The glycogen content was determined according to *Deng et al. (2016)*. The concentration was expressed in mmol of glycosyl residues/g tissue.

### Inflammatory cytokines measurement
The pro- and anti-inflammatory cytokines interleukin IL-1β, IL-2, IL-4, IL-5, IL-6, IL-10, IL-12p70, tumor necrosis factor-α (TNF-α), keratinocyte-derived chemoattractant/human growth-regulated oncogene (KC/GRO) and interferon-γ (IFN-γ) were measured using

Meso Scale multiplex assay kit (V-PLEX, Pro-inflammatory Panel 1 mouse kit, Ref: K15048D-1; Meso Scale, Rockville, MD, USA). Optical density was read using an MSD instrument (SECTOR Imager 2400), and the data were analyzed using Proprietary Meso Scale software.

## Statistical analysis

Statistical analysis was carried out by one way ANOVA followed by Dunnett's Multiple Comparison Test and Student $t$-test using GraphPad Prism software version 5.0. All results are presented as means $\pm$ S.E.M. Statistical significance is indicated by $*P < 0.05$, $**P < 0.01$; $***P < 0.001$.

## RESULTS

### Glycerol release in 3T3-L1 cells

The lipolytic effect of sage in 3T3-L1 cells was determined by measuring glycerol release (Fig. 1A). Compared to the untreated cells, no difference was observed in glycerol levels measured in the medium of treated cells with the plant extract at 0.2, 1, 5, 25 or 50 µg/ml.

### Lipid droplets accumulation

When the cells were treated with sage extract during the differentiation or the nutrition step, (Figs. 1B and 1C) there was a significant reduction in the lipid accumulation only in the cells treated with highest two concentrations (25 and 50 µg/ml). However, when sage extract was added to both differentiation and nutrition media (Fig. 1D), there was an inhibitory effect on lipid droplets accumulation even with the low sage concentration (0.2 and 1 µg/ml) with an overall concentration-dependant manner.

### Oral glucose tolerance test (OGTT) and insulin tolerance test (ITT)

OGTT was performed after 14 days of treatment. There was no change in fasted blood glucose in all treated groups, but 30 min following the glucose load, the blood glucose levels in mice treated with high ($p < 0.05$) and low ($p < 0.001$) dose of sage extract, and mice treated with rosiglitazone ($p < 0.001$) were significantly lower, compared to the control group values (Fig. 2A). Moreover, the low dose of sage extract exhibited a similar effect to that of rosiglitazone on blood glucose, and tended to decrease fasting plasma insulin levels ($p = 0.08$). Compared to the control mice, the plasma insulin levels measured 30 min in response to glucose load, were significantly lower in rosiglitazone treated group ($p < 0.001$), and in low ($p < 0.01$) and high dose ($p < 0.05$) treated groups (Fig. 2B).

After 3 weeks, there was no difference in 5 h fasted blood glucose levels of sage or rosiglitazone treated mice compared to control group (Fig. 3A). However, the treatment with high and low dose sage, and rosiglitazone resulted in a significant reduction in fasted plasma insulin levels (high dose sage: $698 \pm 132$ pmol l$^{-1}$, $p < 0.05$; low dose sage: $491 \pm 50$ pmol l$^{-1}$, $p < 0.01$; rosiglitazone: $284 \pm 36$ pmol l$^{-1}$, $p < 0.001$). Consequently, the insulin sensitivity index, as represented by the Homeostasis Model Assessment of Insulin Resistance (HOMA-IR) was respectively reduced by 39, 60 and 78%, indicating a marked improvement in insulin sensitivity by both sage extract and rosiglitazone (Fig. 3B).

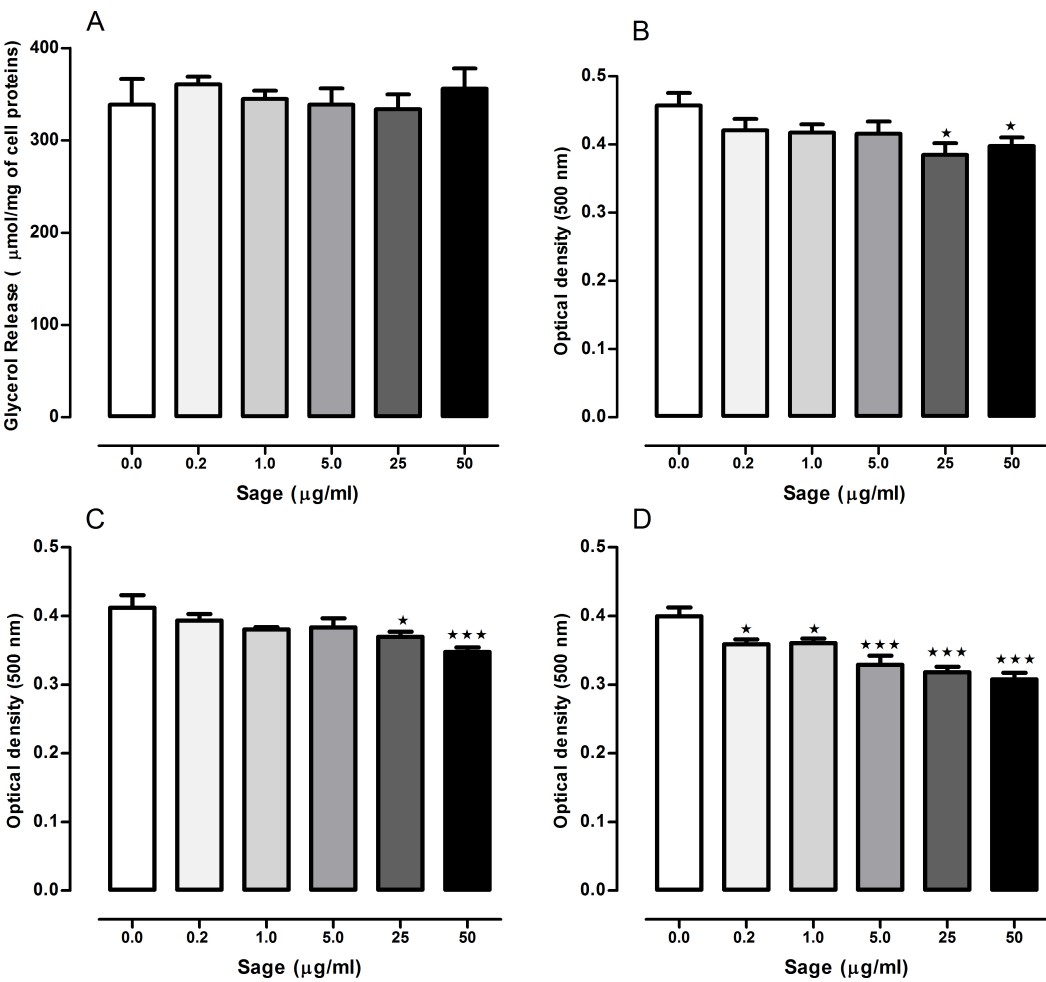

**Figure 1   Effects of *Salvia officinalis* methanol (sage MetOH) extract on lipolysis and lipogenesis in 3T3-L1 cells.** Levels of glycerol released in the culture medium of fully differentiated adipocytes after being incubated for 90 min with or without 0.2, 1, 5, 25 and 50 $\mu$g/ml of methanol sage extract (A). Lipid droplets accumulation measured in 3T3-L1 cells treated with different sage extract concentrations during the stage of differentiation (B), nutrition (C), or both (D). All values are mean ± SEM ($n = 4$, in each treated cell group). Statistical analysis were performed using one way anova test followed by Dunnett's multiple comparison test. *$p < 0.05$; ***$p < 0.001$.

An ITT was carried out after four weeks of treatment. The fasted blood glucose levels were reduced in low dose sage and rosiglitazone groups compared to control ($p < 0.05$ and $p < 0.01$, respectively) (Fig. 3C). Both groups showed a significant drop in blood glucose levels 10 and 20 min following the insulin injection (Fig. 3C). Despite the lower blood glucose levels in response to insulin injection in high dose sage treated group, no statistical significance was observed. Nevertheless, the ITT supported evidence from the OGTT that the sage extract improved insulin sensitivity.

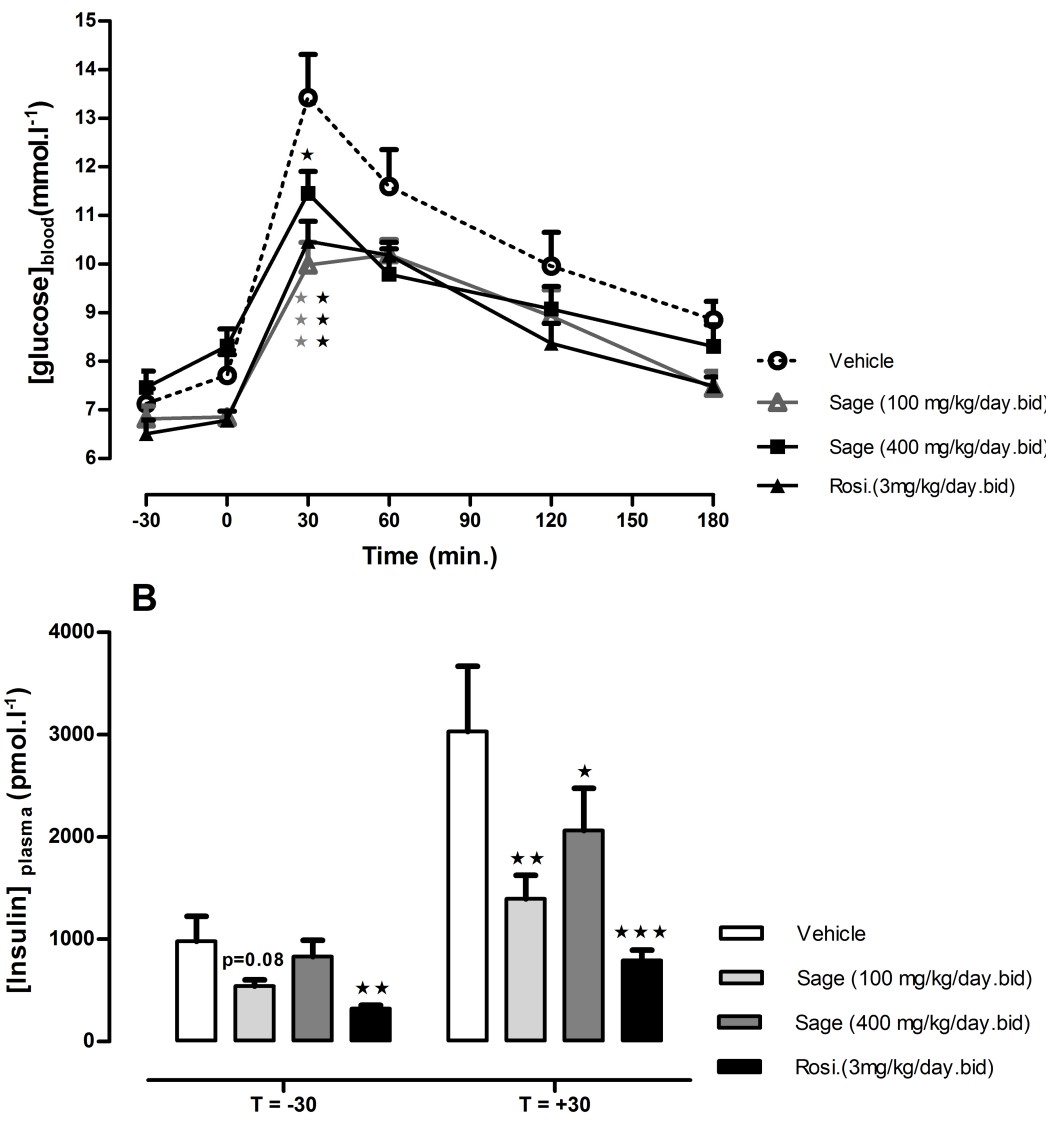

**Figure 2** Oral glucose tolerance test (A) and plasma insulin levels 30 min before and after glucose load (B). HFD mice were treated for two weeks with sage MetOH extract (100 mg/kg/day and 400 mg/kg/day), and rosiglitazone (3 mg/kg/day). Glucose solution (2.5 g/kg) was given orally after 5 h fast. Blood glucose values represent mean + SEM ($n = 7$ in group A and $n = 8$ in groups B, C and D), and statistical significance compared with the vehicle group data is shown as: $*P < 0.05$, $**P < 0.01$ and $***P < 0.001$.

## Effect of treatment on biochemical parameters

After 5 weeks of treatment, there was no change in fed plasma leptin, total cholesterol and HDL-cholesterol levels (Figs. 4B–4D). On the other hand, the low dose of sage extract significantly decreased fed plasma insulin, triglycerides and NEFA levels (Figs. 4A, 4E and 4F). There was no difference in liver triglycerides and glycogen contents between all groups.

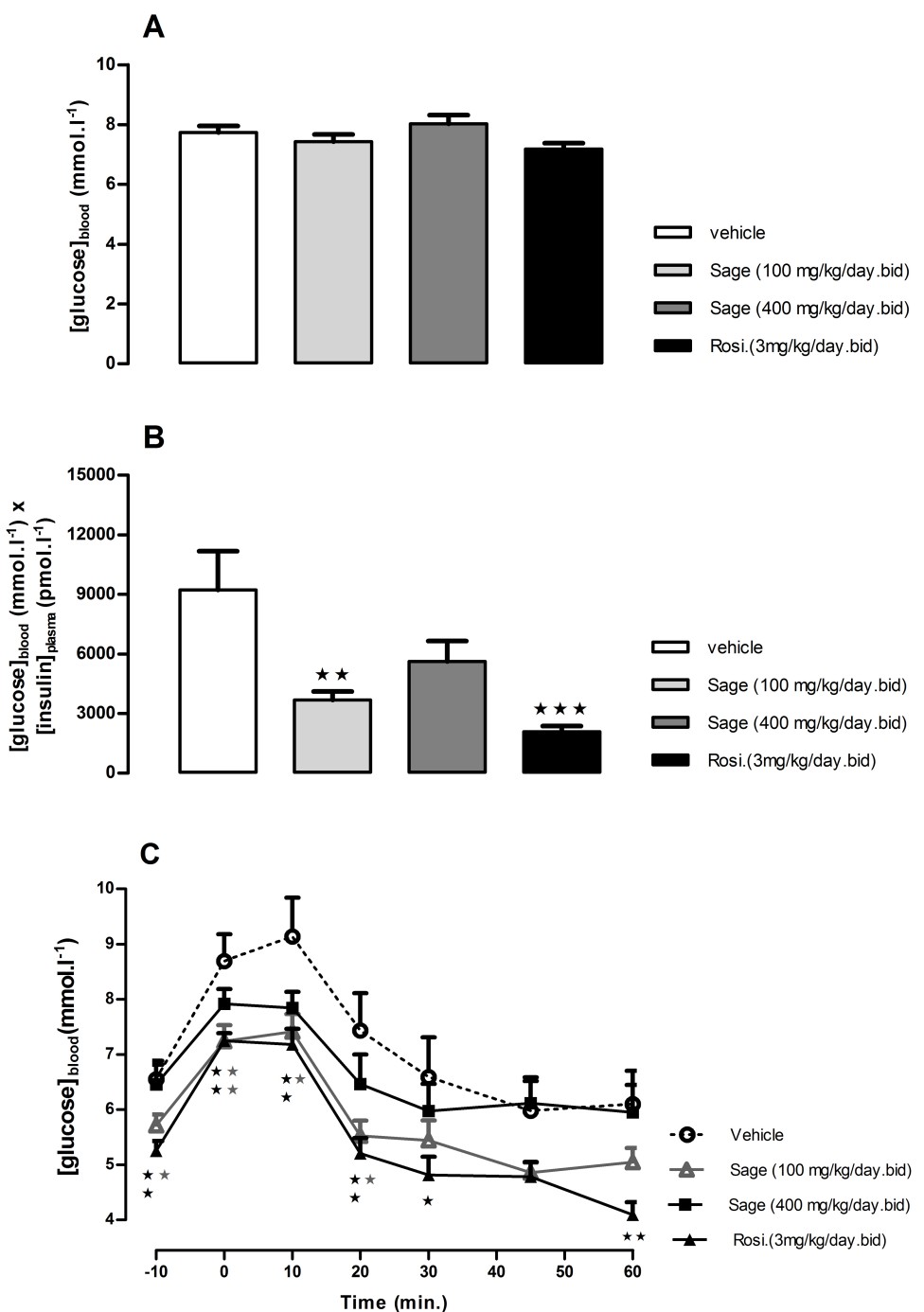

**Figure 3** **Effects of Salvia officinalis MetOH extract on fasted blood glucose, and insulin sensitivity.** Blood glucose levels were measured after a 5-hour fast (A), and insulin sensitivity is represented by the product of blood glucose and plasma insulin values (B), after three weeks of treatment with sage MetOH extract (100 mg/kg/day and 400 mg/kg/day), and rosiglitazone (3 mg/kg/day). Blood glucose levels were observed during the Insulin tolerance test (C), performed at day 29 of treatment. Insulin (Actrapid, 0.75 UI/kg/ml in saline) was injected intraperitoneally to 5-hour fasted mice. All values are means ± SEM ($n = 7$ in group A and $n = 8$ in groups B, C and D). *$P < 0.05$, **$P < 0.01$ and ***$P < 0.001$ as compared to the vehicle-treated group.

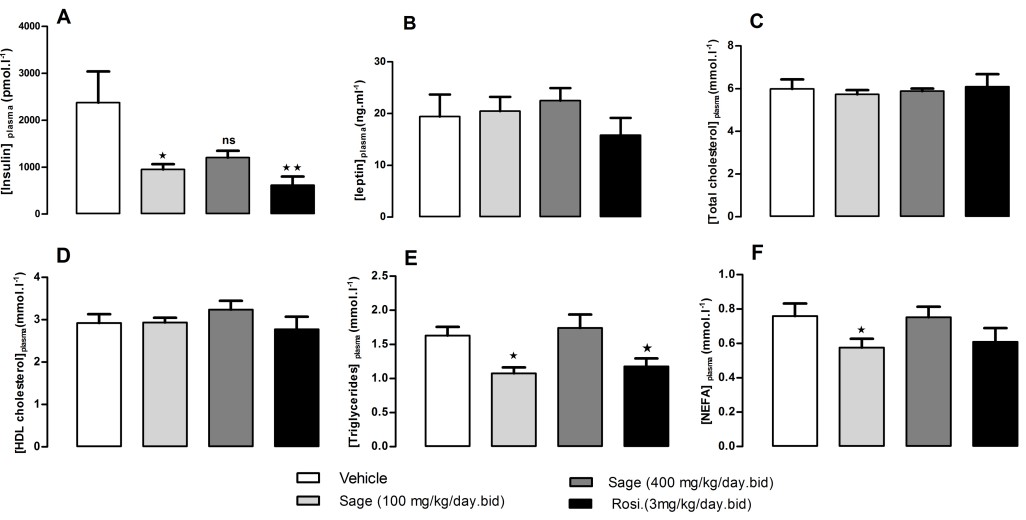

**Figure 4** **Plasma levels of insulin (A), leptin (B), total cholesterol (C), HDL-Cholesterol (D), Triglycerides (E), and NEFA (F), after five weeks of treatment with sage MetOH extract.** The plasma samples were collected from fed mice, at the termination day, after five weeks treatment with sage MetOH extract (100 mg/kg/day and 400 mg/kg/day), and rosiglitazone (3 mg/kg/day). Values represent mean ± SEM ($n = 7$ in group A and $n = 8$ in groups B, C and D), and statistical significance compared with the vehicle group data is shown as: $^*P < 0.05$ and $^{**}P < 0.01$.

**Table 2** **Cumulative food intake, bodyweight change, lean and fat mass at day 32 of treatment.** Lean and fat mass were measured by Nuclear Magnetic Resonance (NMR).

|  | Group A Control | Group B 100 mg kg$^{-1}$ | Group C 400 mg kg$^{-1}$ | Group D Rosiglitazone |
|---|---|---|---|---|
| Cumulative food intake (g) | 73.5 ± 3.6 | 61.5 ± 2.4$^*$ | 68.6 ± 2.3 | 64.7 ± 3.1 |
| Cumulative bodyweight change (g) | 4.51 ± 1.14 | 0.96 ± 0.73$^*$ | 3.39 ± 0.89 | 1.95 ± 0.82 |
| Lean mass (g) | 24.6 ± 2.2 | 23.3 ± 1.4 | 25.5 ± 2.0 | 24.0 ± 1.6 |
| Fat mass (g) | 19.0 ± 0.6 | 16.6 ± 0.5$^*$ | 18.3 ± 0.8 | 17.6 ± 0.6 |

**Notes.**
$^*p < 0.05$ as compared to group A. Data are expressed in mean SEM; ($n = 7$ in group A and $n = 8$ in groups B, C and D).

## Effect of treatment on food intake, on body weight gain and energy expenditure

Cumulative food intake after 34 days of treatment was reduced in group B (100 mg kg$^{-1}$) compared to the other groups (Table 2). Moreover, a decline ($p = 0.018$) in cumulative bodyweight change was observed for the same group compared to the control group (Table 2). This reduction in body weight in group B, could be explained by a decrease ($p = 0.011$) in fat mass without any changes in lean mass (Table 2). Energy expenditure was measured for 24 h after 35 days of treatment. There were no significant effects of treatment on expenditure per animal or relative to body weight over the whole 24 h or during the light or dark phases (data not shown).

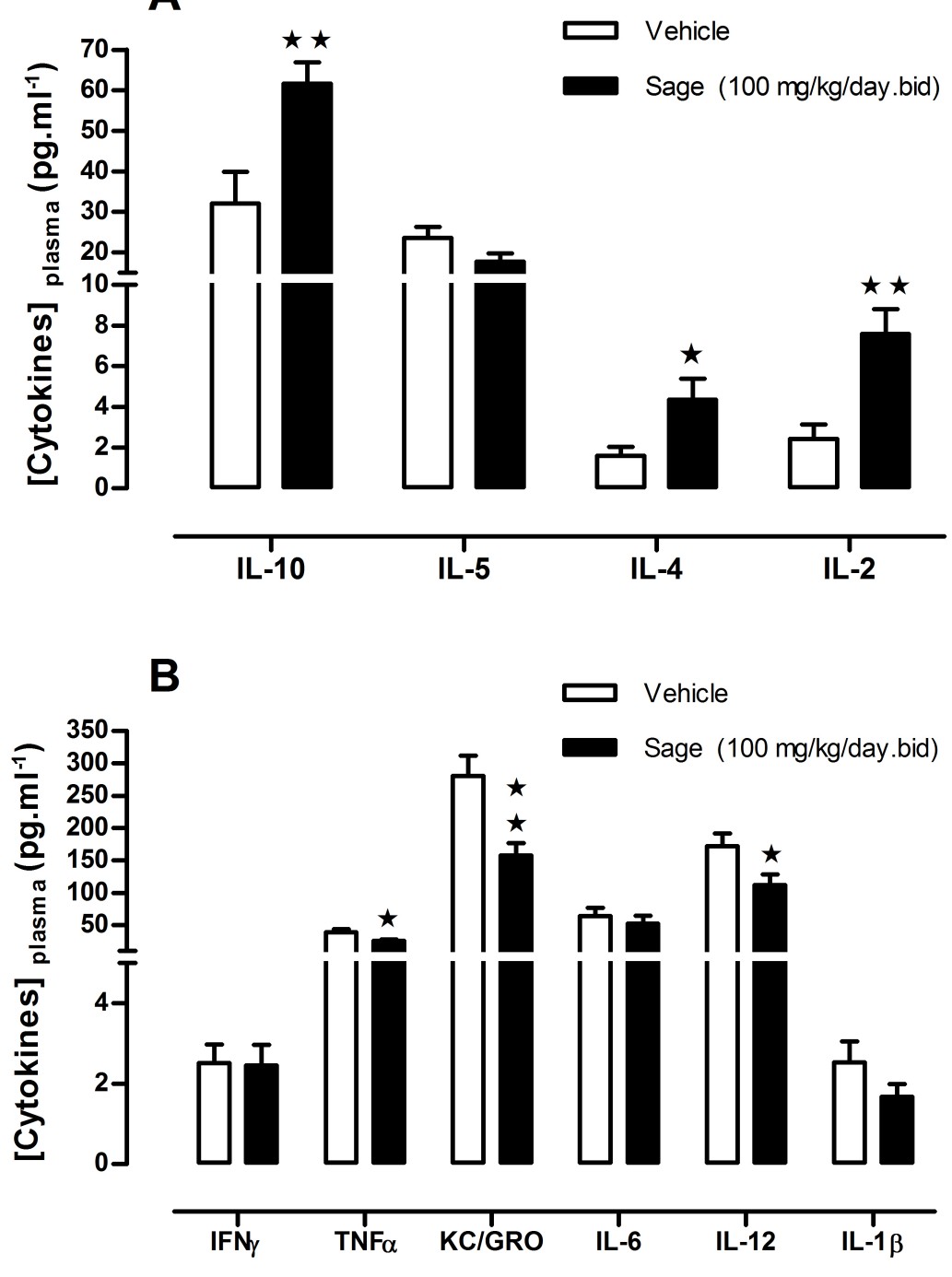

**Figure 5** **Plasma levels of 'anti- and pro- inflammatory' cytokines after chronic treatment with low dose of sage MetOH extract.** Comparison between the levels of anti-inflammatory (A), and pro-inflammatory cytokines (B), measured in plasma samples collected from fed mice treated for five weeks with sage extract low dose sage MetOH extract (100 mg/kg/day), and mice control group (water: 10 ml/kg/day). Values represent mean ± SEM ($n = 7$ in group A and $n = 8$ in group B) , and statistical significance compared with the vehicle group data is shown as: *$P < 0.05$ and **$P < 0.01$.

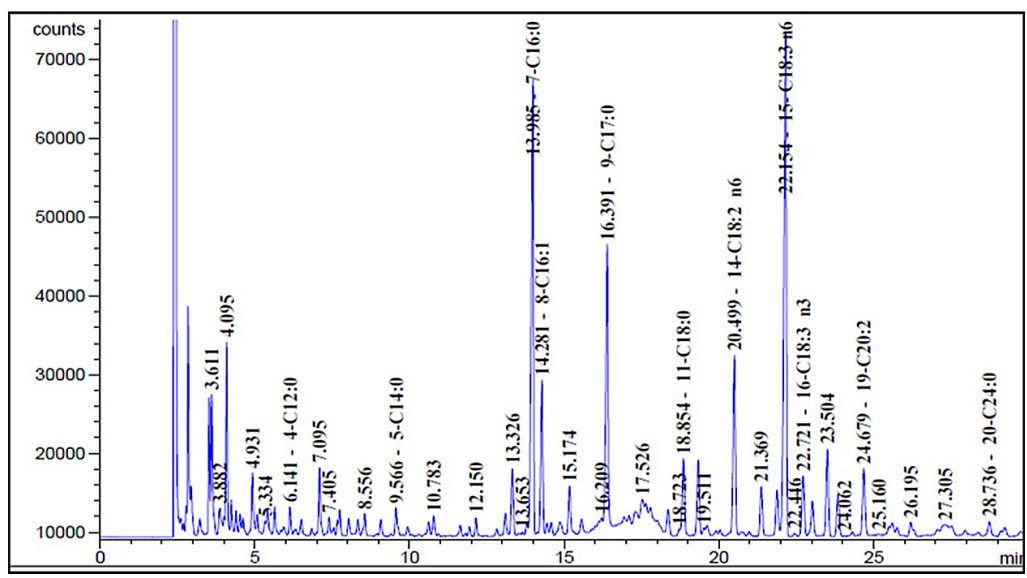

**Figure 6  Sage fatty acids profile analysed by Gas Chromatography (GC).** Typical chromatogram of sage fatty acid profile analyzed by GC. The composition of fatty acid in sage MetOH extract contains different classes ranging from C4:0 to C24:0. FA methyl esters (FAMEs) were identified by comparing each sample with a standard FAME reference mixture, and FA acid peak areas were calculated using HP ChemStation software.

### Pro- and anti-inflammatory cytokines

To investigate a potential anti-inflammatory effect of sage, plasma levels of a panel of pro- and anti-inflammatory cytokines were measured. Compared to the control group, there was an increase in the anti-inflammatory cytokines IL2, IL-4 and L-10 ($p = 0.003$, $p = 0.036$ and $p = 0.006$; respectively) (Fig. 5A), and a decrease in pro-inflammatory cytokines TNF-α, KC/GRO and IL-12 ($p = 0.030$, $p = 0.004$ and $p = 0.034$; respectively), moreover the sage extract tend to reduce as well the plasma levels of IL-1β (Fig. 5B).

### Fatty acids composition of Sage MetOH extract

As displayed in Fig. 6, the sage MetOH extract chromatographic analysis allowed the identification of different FAs classes. The main FAs identified were: γ-linolenic acid (30.51%), palmitic acid (24.31%), linoleic acid (10.41%), palmitoleic acid (7.99%), eicosadienoic acid (4.01%), oleic acid (4.34%) and $\alpha$-linolenic acid (3.49%) (Table 3).

### DISCUSSION

Anti-diabetic properties of sage have been shown in *in vitro* as well as in *in vivo* studies. However, all previous studies using diabetic animal models have been performed in streptozotocin or alloxan mice or rats (*Alarcon-Aguilar et al., 2002*; *Eidi, Eidi & Zamanizadeh, 2005*; *Eidi & Eidi, 2009*). Both streptozotocin and alloxan induce β-cell death through alkylation of DNA by the nitrosourea moiety of these compounds. For this reason, the previous animal models of diabetes are more representative of type 1 diabetes (T1D) than type 2 diabetes (T2D). Type 2 diabetes is more common in older

**Table 3 Fatty acids (FAs) composition of sage Methanol extract, measured by Gas Chromatography (GC).** FAs were analysed by GC in the following conditions: injector and Flame Ionization Detector (FID) temperature were set at 250 °C and 280 °C, respectively. Oven temperature was kept at 15 °C for 1 min then gradually raised to 230 °C at 10 °C/min and subsequently, held isothermal for 10 min. Nitrogen was the carrier gas at a split ratio of 1:50, a linear velocity of 38.5 cm/s and a flow rate of 1.2 ml/min.

|  | Nomenclature | Name | Content (µg/g) | Percentage% |
|---|---|---|---|---|
| Saturated fatty acids (SFAs) | C4:0 | Butyric acid | 27.99 | 0.98 |
|  | C6:0 | Caproic acid | 28.11 | 0.99 |
|  | C10:0 | Citric acid | 26.94 | 0.95 |
|  | C12:0 | Lauric acid | 35.14 | 1.23 |
|  | C14:0 | Myristic acid | 44.27 | 1.55 |
|  | C16:0 | Palmitic acid | 692.34 | 24.32 |
|  | C18:0 | Stearic acid | 127.67 | 4.48 |
|  | C20:0 | Arachidic acid | 71.30 | 2.50 |
|  | C24:0 | Lignoceric acid | 32.40 | 1.14 |
| Monounsaturated fatty acids (MUFAs) | C16:1 ω-7 | Palmitoleic acid | 227.71 | 8.00 |
|  | C18:1 ω-9 | Oleic acid | 123.70 | 4.34 |
|  | C18:1 ω-7 | Vaccenic Acid | 22.89 | 0.80 |
|  | C20:1 ω-9 | Gadoleic acid | 7.36 | 0.26 |
|  | C22 1 ω-9 | Erucic acid | 0.2 | 0.01 |
|  | C24:1 ω-9 | Nervonic acid | 0.12 | <0.01 |
| Polyunsaturated fatty acids (PUFAs) | C18:2 ω-6 | Linoleic acid | 296.40 | 10.41 |
|  | C18:3 ω-6 | γ-linolenic acid | 868.69 | 30.51 |
|  | C18:3 ω-3 | α-linolenic acid | 99.48 | 3.49 |
|  | C20:2 ω-6 | Eicosadienoic acid | 114.29 | 4.01 |

male population and associated with persistent low grade inflammation related to adipose tissue expansion (*Griffin et al., 2016*; *Pirola & Candido Ferraz, 2017*). Hence, HFD-induced obesity in 15 to 16 weeks old male mice presents an excellent model of the pre-T2D state, since the prevalence of T2D is greatly increased in obesity. The present study investigated the effect of sage extract on lipolysis and lipogenesis in murine pre-adipocytes (3T3-L1), as well as the protective properties of low and high doses of the plant extract on inflammation, obesity and insulin resistance in a HFD animal model.

Our finding *in vitro* revealed that sage MetOH extract has no effect on lipolytic activity in 3T3-L1 cells; however *s*age-treated animals had a decrease in plasma NEFA and triglycerides levels, suggesting an inhibition of lipolysis. This may be because 3T3-L1 cells differ from primary adipocytes, or because *in vivo* there are influences of sage on other tissues that indirectly lower plasma NEFA or triglyceride levels. In sage treated mice, the plant extract improves insulin sensitivity, this improvement may be partly due to a direct effect on adipose tissue and thereby it may enhances the anti-lipolytic effect of insulin, resulting in a decrease in plasma NEFA levels. Furthermore, lipolysis in adipose tissue is modulated *in vivo* by various factors such as insulin, glucagon, epinephrine, norepinephrine, ghrelin, growth hormone, testosterone, and cortisol, which is not the case *in vitro*.

In addition, the plant extract reduced lipid droplets accumulation in a concentration-response manner when the cells were treated during both the differentiation and nutrition

steps. Obesity is a pathological disorder characterized by excessive fat storage endogenously; and here we demonstrate that sage MetOH extract exhibited significant *in vitro* inhibition of lipid droplets accumulation in mature adipocytes. However, sage extract was more efficient on reducing lipid storage when it was added to both differentiation and nutrition media (Fig. 1D) instead to differentiation or nutrition medium alone (Figs. 1B and 1C). Further analysis of the upregulation or downregulation of genes such as CCAAT/enhancer binding protein-α (C/EBP-α) and peroxisome proliferator-activated receptor gamma (PPAR-γ), involved in the differentiation of pre-adipocytes into fully mature adipocytes, will indicate which mechanisms sage employs in decreasing lipogenesis.

The efficacy of *S. officinalis* in the prevention of lipid accumulation within 3T3-L1 adipocytes is consistent with it reducing bodyweight gain (related to adipose tissue hypertrophy) in HFD mice treated with low dose sage MetOH extract. Nevertheless, this anti-obesity effect of sage might be due also to reduced food intake (Table 2).

In HFD animals, the treatment with low dose MetOH extract for two weeks, resulted in an improvement of glucose tolerance, and a reduction of plasma insulin levels in response to glucose load. After three weeks of treatment, the HOMA-IR index reflecting insulin resistance was significantly decreased in sage treated-animal. The results confirm an improvement in tissue insulin sensitivity. This result is in contrast with those of *Eidi & Eidi (2009)*, who reported that after two weeks sage ethanol extract significantly decreased serum glucose, whereas it increased serum insulin levels in treated induced diabetic rats by STZ as compared with control diabetic rats, and those of *Alarcon-Aguilar et al. (2002)* who demonstrated that the water ethanol extract of sage showed hypoglycemic activity in both normo-glycemic and in mildly alloxan-diabetic mice, but required the presence of insulin to exhibit its activity. On the other hand, *Eidi, Eidi & Zamanizadeh (2005)* reported same results as our study, showing that intraperitoneal administration sage MetOH extract significantly decreased blood glucose in fasted STZ-diabetic rats without increasing insulin release.

Our evidence from blood glucose and plasma insulin levels that low dose sage improved insulin sensitivity was confirmed by an insulin tolerance test conducted after four weeks of treatment. Thus, mice treated with sage low dose exhibited a significant decrease in blood glucose levels in response to intraperitoneal insulin injection. This test, to the best of our knowledge, was performed on sage-treated mice for the first time and confirms an insulin-sensitizer effect of sage extract. Moreover the chronic treatment with low sage dose resulted in a significant decrease in fed plasma insulin levels. A similar effect was observed in the rosiglitazone-treated mice.

Sage, at low dose ($100 \text{ mg kg}^{-1}$/day), has shown a better beneficial effects on glucose tolerance, insulin sensitivity, bodyweight gain and food intake, when compared with high dose ($400 \text{ mg kg}^{-1}$/day). This might be explained by an increased daily intake in palmitic acid (PA). According to the content in PA of sage MetOH extract (Table 3), the daily intake of PA, in high dose sage treated mice was $277 \text{ μg kg}^{-1}$, compared to $69.2 \text{ μg kg}^{-1}$ in low dose treated group. High levels of PA lead to insulin resistance (*Reynoso, Salgado & Calderón, 2003*).

Indeed, saturated fatty acids are major contributors to this process, as they directly impair insulin sensitivity in adipocytes and muscle cells in culture through pro-inflammatory effect-induced insulin resistance (*Bilan et al., 2009*). Moreover, in human study, *Stevenson, Clevenger & Cooper (2015)* have reported that PUFAs (*Bilan et al., 2009*) rich diet consumption resulted in a decrease in ghrelin and an increase in peptide YY plasma levels, when compared to SFAs rich diet. This might explain the beneficial effect on food intake of sage low dose when compared to high dose. Nevertheless, it is reasonable to explain the absence of the beneficial effects of high dose by a possible presence of compounds that could be toxic at high levels. However, *Eidi, Eidi & Zamanizadeh (2005)* have used higher dose (500 mg/kg/day) and no toxic effect has been observed. Moreover, *Alves Rodrigues et al. (2012)*, have investigated in mice the toxicological effects of the hydroalcoholic extract from *Salvia officinalis* leaves. In the acute toxicity test, the LD50 value was close to 45 g/kg, which is over than 110 fold when compared to the high dose (400 mg/kg/day) used in our study. The HPLC profile (Table 4) of sage methanol extract showed that sage contains 13 compounds with almost no presence of toxic compounds. We have also conducted a preliminary study on dose–response of methanol sage extract on 3T3-L1 cell proliferation and cell viability (using methylene blue assay), no toxic effect has been observed.

This finding suggests that sage MetOH extract might act more efficiently at low doses, on bodyweight gain and insulin resistance. Further studies using lower doses at 50, 25 and 10 mg kg$^{-1}$ are needed to select the most effective dose. Adding palmitic acid to the dose extract should also be taking into consideration to see whether it prevents the beneficial effect.

*Alves Rodrigues et al. (2012)*, showed that oral administration of the hydroalcoholic extract and active compounds isolated from sage, such as carnosol, oleanolic and ursolic acids reduced the nociception and oedema induced by different chemical stimuli. A study showed that *S. officinalis* may be used just as an adjuvant in anti-inflammatory therapy (*Oniga et al., 2007*). Anti-inflammatory activity of sage essential oil assessed *in vitro* showed that it significantly inhibited nitric oxide production elicited by LPS in macrophages (*Abu-Darwish et al., 2013*). Our study evaluated, for the first time, sage MetOH leaf extract effect on plasma inflammatory cytokines in HFD animals. Sage increased significantly the plasma levels of anti-inflammatory cytokines (IL-2, IL-4 and IL-10) and exhibited an opposite effect on pro-inflammatory cytokines by decreasing the plasma levels of TNF-α, KC/GRO and IL-12. KC/GRO is highly induced by pro-inflammatory cytokines such as TNF-α (*Son et al., 2007*) and the fact that plasma levels of TNF-α were decreased ($p = 0.030$) suggests that sage extract modulates cytokines gene expression by down regulating TNF-α expression and indirectly inhibiting KC/GRO release. The beneficial ant-inflammatory effects of sage methanol extract might be related to the high levels of phenolic compounds (Table 4), particularly in Rosmarinic acid (*Jin et al., 2017*). Nevertheless, a study on the expression of IL-10, IL-4, TNF-α and KC/GRO in white adipose tissue is needed to confirm the mechanism of action.

PUFAs prevent diet-induced insulin resistance in rodents (*Storlien et al., 1987*). Indeed, various FAs serve as natural ligands for the three subtypes of PPARs: α, γ and δ (*Christensen et al., 2009*). PPAR α and γ agonists modulate important metabolic events and they are the

**Table 4  Phenolic compounds content of _S. officinalis_ methanol extract.** The phenolic compounds separation was performed using a Hewlett-Packard liquid chromatographic system. The identification and quantification of compounds was carried out by comparison of their retention times and pics area, with respect to pure standards analyzed under the same operating conditions. Values are given as mean ± standard error of three replicate analyses.

| | Phenolic compounds | $R_T^a$ (min) | Content (mg/Kg of DW)[b] |
|---|---|---|---|
| | **Phenolic acid** | | |
| 1 | Gallic acid | 5.029 | 11.08 ± 1.2 |
| 2 | 4-OH Benzoic acid | 7.559 | 9.95 ± 1.3 |
| 3 | Rosmarinic acid | 12.458 | 290.73 ± 8.4 |
| 4 | Cinnamic acid | 17.138 | 17.13 ± 2.6 |
| | **Flavones** | | |
| 5 | Luteolin-7-_O_-glucoside | 9.230 | 10.89 ± 1.7 |
| 6 | Apigenin-7-_O_-glucoside | 10.616 | 10.80 ± 1.8 |
| 7 | Apigenin | 16.700 | 18.50 ± 2.2 |
| | **Flavonols** | | |
| 8 | Rutin | 8.429 | 10.15 ± 1.5 |
| 9 | Quercetin | 15.101 | 2.26 ± 0.4 |
| | **Stilbenes** | | |
| 10 | Resveratrol | 15.411 | 7.15 ± 1.2 |
| | **Catechins** | | |
| 11 | Catechin Hydrate | 6.278 | 6.75 ± 0.9 |
| | **Phenylethanoid glycoside** | | |
| 12 | Verbascoside | 8.806 | 17.47 ± 1.6 |
| | **Lignans** | | |
| 13 | Pinoresinol | 15.782 | 1.84 ± 0.3 |

**Notes.**
[a] Retention time.
[b] Dry weight.

targets of drugs or candidate drugs that are effective in the treatment of metabolic disorders such as T2D, atherosclerosis (_Berger, Akiyama & Meinke, 2005_) and obesity (_Bassaganya-Riera, Guri & Hontecillas, 2011_). Natural PPARs have fewer adverse effects than novel synthetic PPARs ligands that are suspected to promote carcinogenesis in rodents by as yet, unknown mechanisms (_Berger, Akiyama & Meinke, 2005_). Nevertheless, other studies have reported an antitumor effects of the PPAR-γ agonists thiazolidinediones (TZDs), _in vivo_ (_Chaffer et al., 2006_), as well as _in vitro_ (_Xu et al., 2003_). These anti-proliferative effects of TZDs, including cell cycle arrest and/or increased apoptosis, are PPAR-γ-pathways dependent in some cancer cell types, while in other tumor cells, they occur independently.

GC analysis showed that sage MetOH extract had a high content in PUFAs (48.4%), particularly in γ-linolenic acid, linoleic acid, and α-linolenic acid. PPAR-γ is effectively activated by PUFAs, such as linolenic acid (_Dubuquoy et al., 2002_). Indeed, γ-linolenic and α-linolenic acids alone account for 34% of sage FAs content in MetOH extract, and both of them are PPAR-α and γ agonists (_Christensen et al., 2010_; _Xu et al., 1999_). Furthermore, several studies reported that the PUFA linoleic acid can function as ligands for both PPAR-α and PPAR-γ (_Kliewer et al., 1997_).

Our finding revealed the presence of both PPAR α and γ agonists in MetOH sage extract and support previous studies which demonstrated that dichloromethane (DCM) and Ethanol (EtOH) sage extracts contains PPARs agonists such as α-linolenic acid, γ-linolenic acid, carnosic acid, oleanolic acid, ursolic acid and carnasol (*Christensen et al., 2010*; *Lim et al., 2007*; *Rau et al., 2006*).

## CONCLUSION

Sage MetOH extract shows an anti-adipogenic effect by *in vitro* inhibition of lipid droplets accumulation in adipocytes. In a nutritional model obesity associated with insulin resistance, sage MetOH extract reduces bodyweight gain by a decrease in total fat mass and exhibits anti-diabetic properties by an improvement of glucose tolerance and insulin sensitivity. Sage MetOH extract moderately improves lipid profile, also reduces the plasma levels of the pro-inflammatory cytokines TNF-α, KC/GRO and IL-12 and increases the anti-inflammatory cytokines IL-2, IL-4 and IL-10. Our results suggest that decreased adipose tissue associated with improved insulin sensitivity and modulation of inflammatory cytokines release, balance the abnormal glucose metabolism observed in a pre-diabetic state.

## ACKNOWLEDGEMENTS

The authors dedicate this work to the memory of Professor Michael A. Cawthorne previous director of Buckingham Institute for Translational Medicine, who passed away in July 2015.

### Funding

This work was supported by Buckingham Institute for Translational Medicine (BITM), based in the Clore Laboratory at the University of Buckingham, United Kingdom. Part of this work has been supported by the ''Tunisian Ministry of Higher Education, Scientific Research and Technology''. The funders had no role in study design, data collection and analysis, decision to publish, or preparation of the manuscript.

### Grant Disclosures

The following grant information was disclosed by the authors:
Buckingham Institute for Translational Medicine (BITM).
Tunisian Ministry of Higher Education, Scientific Research and Technology.

### Competing Interests

The authors declare there are no competing interests.

### Author Contributions

- Mohamed R. Ben Khedher conceived and designed the experiments, performed the experiments, analyzed the data, contributed reagents/materials/analysis tools, wrote the paper.

- Mohamed Hammami conceived and designed the experiments, reviewed drafts of the paper.
- Jonathan R.S. Arch analyzed the data, reviewed drafts of the paper.
- David C. Hislop and Dominic Eze performed the experiments, analyzed the data, prepared figures and/or tables.
- Edward T. Wargent and Małgorzata A. Kępczyńska performed the experiments, reviewed drafts of the paper.
- Mohamed S. Zaibi conceived and designed the experiments, performed the experiments, analyzed the data, contributed reagents/materials/analysis tools, wrote the paper, prepared figures and/or tables, reviewed drafts of the paper.

## Animal Ethics

The following information was supplied relating to ethical approvals (i.e., approving body and any reference numbers):

Animal experiments were conducted in accordance with ethical procedures and policies approved by the UK Government Animal Act 1986 (Scientific procedures) and Animal Welfare and Ethical Review Board (AWERB) of the University of Buckingham, UK.

## Data Availability

The raw data has been provided as Data S1.

## Supplemental Information

Supplemental information for this article can be found online at http://dx.doi.org/10.7717/peerj.4166#supplemental-information.

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
