# Peer review of "Preventive effects of Salvia officinalis leaf extract on insulin resistance and inflammation in a model of high fat diet-induced obesity in mice that responds to rosiglitazone"

_PeerJ, doi:10.7717/peerj.4166_

## Round 0.1 · original submission · Major Revisions

Please, address each and every point raised by the reviewers in a revised manuscript.

Reviewer 1 ·

Basic reporting

MR. Ben Khedher and colleagues investigates whether Salvia officinalis (Sage) leaf extract prevent insulin resistance and inflammation in high fat diet-induced-obesity mice model. It should be emphasized that 598 papers (see Pub Med) were written about the sage effects. Among these references, it is described that sage exerts hypoglycaemic and anti-dyslipidemic effects in diabetic subjects (https://www.ncbi.nlm.nih.gov/pubmed/27942500; https://www.ncbi.nlm.nih.gov/pubmed/25340127; https://www.ncbi.nlm.nih.gov/pubmed/21506190).
This study may be considered original (inflammatory state related adipose tissue) and may follow a publication. In spite of that, some ambiguities need to be clarified.

Problematic question: Why the authors chose to compare the sage effects to rosiglitazone properties, whereas this oral anti-diabetic molecule (Avandia©) has been severely criticized due to cardiac complications leading many deaths, such as myocardial infarction and then withdrawn from the market since 2002. The prestigious British journal NEJM evaluated the deleterious effects of rosiglitazone (https://www.ncbi.nlm.nih.gov/pubmed/17517853). Thanks to argue this aspect.

Relevant question: Sage can also leads to cardiovascular complications effects like rosiglitazone? The authors did not investigate this point in their study (effect on blood pressure?) and no discuss in discussion section. Indeed, in conclusion, the authors confirm that: at low dose, Sage exhibits similar effects to rosiglitazone? Thanks to argue this aspect.

In addition, it is recommended that Rosiglitazone appear in the article-title. For example: “Preventive effects of salvia officinalis leaf extract versus Rosiglitazone on insulin resistance and inflammation, in high fat diet-induced-obesity mice model”

Experimental design

Diet: 60% fat by energy value is no sufficient. What is the ratio of diet lipid composition to saturated, monounsaturated and polyunsaturated fatty acids, respectively? Also, the composition to others nutriments: proteins, carbohydrates, vitamins and minerals?

Fasting time: 5 hours is insufficient to study metabolism. 16 hours minimum. What protocol and reference?
Insulin resistance (IR): The protocol used is not clear. Why to use Oral Glucose Tolerance Test and the Insulin Tolerance Test separately? In practice (experimental animal but not in human), IR is evaluated by the hyperinsulinemic euglycemic clamp. HOMA model is no sufficient. Please clarify this point.

Obesity mice model or diabetic mice model? As the authors studied rosiglitazone (oral antidiabetic), It is important to confirm the mice C57Bl6 (transgenic model) is also a model of type 2 diabetes, and to be associated with insulin resistance. The authors have not studied. Please clarify this point.

Statistical analysis: The correlation test (Pearson test) between the comparative effects between sage and rosiglitazone is missing. Student t-test is no sufficient.

Validity of the findings

In vitro versus in vivo: The results are obtained from both the pre-adipocytes cells (3T3-L1) and mice (in vivo study). What relationship between these two studies, since the cells were not isolated from the obesity model mice (C57Bl6)?

Anthropometric parameters: This study concerns obesity related to glucose tolerance. Although in Table 1 the authors present the weight variations, Results not show whether sage extract effect on Body mass index variation. In addition, the imaging data (Lean and fat mass were measured by Nuclear Magnetic Resonance) for the assessment of fat qualitative mass composition (subcutaneous or visceral) have not been shown.

Lipogenesis versus lipolysis: There is confusion in the data to explain the lipids metabolism effects of sage. The authors confirm that sage exerts both lipogenic and anti-lipolytic effects at the same time (Line 267:... however sage-treated animals had a decrease in plasma NEFA and triglycerides levels, suggesting an inhibition of lipolysis…. Line 271:... the plant extract reduced lipogenesis…). Please clarify this point.

Sage inefficacity or sage toxicity: Line 317:... This might explain the beneficial effect on food intake of sage low dose when compared to high dose…
It’s not clear the dose effect of sage. If sage has no beneficial effect at high doses, can to become toxic? In this case, how will sage be correlated with rosiglitazone? Please clarify this point.

Additional comments

Thank you to see critical above.

In Figure 5, authors should correct that panel A: is anti-inflammatory cytokines and panel B: is pro-inflammatory cytokines.

·

Basic reporting

The authors report positive effects of methanol extracts of Sage (salvia officinalis) on inflammation, obesity and diabetes related parameters both in vitro in 3T3L1 adipocyte differentiation model and in vivo in a high fat nutritional obesity mouse model. The authors appropriately conducted tests related to obesity and diabetic related studies that include glucose and insulin tolerance test and further evaluate the effects of Sage extract on adipokine leptin and inflammatory proteins. Although the in vitro data from differentiated 3T3L1 does not completely correlate with the in vivo data regarding the lipolytic effects of Sage, the authors successfully demonstrate positive effects in both models. In general, the manuscript is sound in logic and writing. The conclusions support the results. Appropriate statistical tests have been performed. Please refer to general comments for the reviewer’s critique.

Experimental design

No comments

Validity of the findings

No comments

Additional comments

1) Since the Sage extract contain several active compounds including toxic compound like alpha and beta-thujones, have the authors conducted a toxicity profile of the methanol extract of Sage. I would recommend a annexin-PI staining of the 3T3L1 cells and a TUNEL staining of frozen section of liver tissue after treatment with SAGE.
2) The authors claim that the presence of several ppar gamma and alpha agonists in the SAGE extracts may be responsible for the observed effects. I would recommend the authors to do a selective knockdown of PPAR gamma or alpha in 3T3L1 adipocytes and/or use the adipose specific knockout mouse model for the PPAR receptor gamma or alpha to prove the validity of results. Alternatively, in addition to the above suggested experiments, can the authors selectively treat cells with ppar gamma or alpha agonists observed in the Sage extract with mass spectrometry data alone or in combinations in vitro to demonstrate the validity of their results.
3) The reviewer would recommend the authors to represent all the in vivo bar graph data for ELISA as scattered or aligned dot plots.

Reviewer 3 ·

Basic reporting

This is a well written manuscript that addresses the effect of sage extract on insulin sensitivity. Sufficient literature references and background context has been provided.

Experimental design

How was the differentiation of adipocytes confirmed? Were adipocytes markers evaluated in the differentiated cells to validate the differentiation? Also, the markers should be tested when differentiation was carried in presence or absence of the extract.

The authors have used two dosing concentrations of sage extract? How were these doses finalized? Also are these doses the weight of the total extract?

What kind of validation was performed on the extract to check for the reproducibility of the contents and concentrations of the component of the extract?

More details about the change in body weight from baseline before treatment to end of the treatment should be presented as a validation of the HFD.

The authors have performed a post-hoc dunnett’s multiple comparison tests that compares the means of the treatment group to the control group. Maybe the authors should consider doing something like Tukey’s to compare all the groups so that they can also judge the difference in effects of the two treatment concentrations.

Validity of the findings

The authors need to touch upon various aspects of HFD model in their discussion to tackle the drawbacks. Male mice are more susceptible to hyperglycemia and inflammation compared to female mice. Further more the authors purport that their study has an advantage over the previous studies since they have used a HFD model vs drug induced diabetes. However they also need to address issues like T2D is a more age associated disease and is more common in the older population. Studies hence caution the use of younger mice.

---

## Round 0.2 · Minor Revisions

The authors have improved their manuscript according to the reviewers' suggestions. However, there are still some points that should be addressed, and specifically:

1) as pointed out by reviewers 1 and 2, sage toxicity profile should be better addressed.
2) as mentioned by reviewer 1, the choice of the fasting times used for dynamic tests in animals should be discussed - the addition of related references may be helpful.
3) in line 360, the authors indicate a role of PPAR agonists in carcinogenesis, although it should be contextually mentioned and cited that TZDs have also anti-proliferative effects, as, for example, indicated by in vitro studies with the insulin receptor gene.
4) lines 267-271: reviewer 1 suggests to better address sage-induced lipogenesis and anti-lipolytic effects.

Reviewer 1 ·

Basic reporting

1. BASIC REPORTING

MR. Ben Khedher and colleagues investigates whether Salvia officinalis (Sage) leaf extract prevent insulin resistance and inflammation in high fat diet-induced-obesity mice model. It should be emphasized that 598 papers (see Pub Med) were written about the sage effects. Among these references, it is described that sage exerts hypoglycaemic and anti-dyslipidemic effects in diabetic subjects (https://www.ncbi.nlm.nih.gov/pubmed/27942500; https://www.ncbi.nlm.nih.gov/pubmed/25340127; https://www.ncbi.nlm.nih.gov/pubmed/21506190).
This study may be considered original (inflammatory state related adipose tissue) and may follow a publication. In spite of that, some ambiguities need to be clarified.

Problematic question: Why the authors chose to compare the sage effects to rosiglitazone properties, whereas this oral anti-diabetic molecule (Avandia©) has been severely criticized due to cardiac complications leading many deaths, such as myocardial infarction and then withdrawn from the market since 2002. The prestigious British journal NEJM evaluated the deleterious effects of rosiglitazone (https://www.ncbi.nlm.nih.gov/pubmed/17517853). Thanks to argue this aspect.

Relevant question: Sage can also leads to cardiovascular complications effects like rosiglitazone? The authors did not investigate this point in their study (effect on blood pressure?) and no discuss in discussion section. Indeed, in conclusion, the authors confirm that: at low dose, Sage exhibits similar effects to rosiglitazone? Thanks to argue this aspect.

In addition, it is recommended that Rosiglitazone appear in the article-title. For example: “Preventive effects of salvia officinalis leaf extract versus Rosiglitazone on insulin resistance and inflammation, in high fat diet-induced-obesity mice model”

Experimental design

Diet: 60% fat by energy value is no sufficient. What is the ratio of diet lipid composition to saturated, monounsaturated and polyunsaturated fatty acids, respectively? Also, the composition to others nutriments: proteins, carbohydrates, vitamins and minerals?

Fasting time: 5 hours is insufficient to study metabolism. 16 hours minimum. What protocol and reference?
Insulin resistance (IR): The protocol used is not clear. Why to use Oral Glucose Tolerance Test and the Insulin Tolerance Test separately? In practice (experimental animal but not in human), IR is evaluated by the hyperinsulinemic euglycemic clamp. HOMA model is no sufficient. Please clarify this point.

Obesity mice model or diabetic mice model? As the authors studied rosiglitazone (oral antidiabetic), It is important to confirm the mice C57Bl6 (transgenic model) is also a model of type 2 diabetes, and to be associated with insulin resistance. The authors have not studied. Please clarify this point.

Statistical analysis: The correlation test (Pearson test) between the comparative effects between sage and rosiglitazone is missing. Student t-test is no sufficient.

Validity of the findings

In vitro versus in vivo: The results are obtained from both the pre-adipocytes cells (3T3-L1) and mice (in vivo study). What relationship between these two studies, since the cells were not isolated from the obesity model mice (C57Bl6)?

Anthropometric parameters: This study concerns obesity related to glucose tolerance. Although in Table 1 the authors present the weight variations, Results not show whether sage extract effect on Body mass index variation. In addition, the imaging data (Lean and fat mass were measured by Nuclear Magnetic Resonance) for the assessment of fat qualitative mass composition (subcutaneous or visceral) have not been shown.

Lipogenesis versus lipolysis: There is confusion in the data to explain the lipids metabolism effects of sage. The authors confirm that sage exerts both lipogenic and anti-lipolytic effects at the same time (Line 267:... however sage-treated animals had a decrease in plasma NEFA and triglycerides levels, suggesting an inhibition of lipolysis…. Line 271:... the plant extract reduced lipogenesis…). Please clarify this point.

Sage inefficacity or sage toxicity: Line 317:... This might explain the beneficial effect on food intake of sage low dose when compared to high dose…
It’s not clear the dose effect of sage. If sage has no beneficial effect at high doses, can to become toxic? In this case, how will sage be correlated with rosiglitazone? Please clarify this point.

Additional comments

Thank you to see critical above

·

Basic reporting

This reviewers comments have been adequately addressed. However the reviewer would appreciate that the authors show a toxicity profile of SAGE extract with a more robust assay like annexin v-PI staning.

Experimental design

No comments.

Validity of the findings

No comments.

Additional comments

Acceptable revision. No further comments.

---

## Round 0.3 · accepted · Accept

The authors have satisfactorily addressed the issues raised by the reviewers.